# Toward an Integrated Model for Soft-Mobility

**DOI:** 10.3390/ijerph16193669

**Published:** 2019-09-29

**Authors:** David Chapman, Agneta Larsson

**Affiliations:** 1Architecture Group, Luleå University of Technology, 97187 Luleå, Sweden; 2Department of Health Sciences, Luleå University of Technology, 97187 Luleå, Sweden; agneta.larsson@ltu.se

**Keywords:** urban design, outdoor activity, health outcomes, climate change

## Abstract

A key urban design challenge is to create built environments that encourage outdoor activity all year round. This study explores a new model for soft-mobility that places the interaction between the urban form, the seasonal climate and climate change, and the individual at the center of people’s soft-mobility choices, or in more general, their modal choice. The research methods used were comparative studies of documents, surveys, mental mapping, and photo elicitation. These studies were undertaken to research people’s outdoor activity in the built environment during the winter season of a cold climate settlement. The results were analyzed against the three-dimensions of the model. In the discussion it is argued that in places with significant climate variation, the interaction between the urban form, the season, and the individual together influence soft-mobility choices. In turn, these interactions influence people’s level of outdoor activity and the individual health benefits such activity can afford. In conclusion, it is highlighted that all three dimensions of the model are in a constant state of change and evolution, especially in relation to planning and development processes and climate change.

## 1. Introduction

In the context of sustainable urban development, good urban design is design that enhances opportunities to be outdoors in the public realm. Here, the weather is a major determinant of people’s decision-making about outdoor activities [1,2]. Therefore, a key premise of this work is that the built environment and the climate play central roles in determining how people use the outdoor environment and the public realm.

Outdoor activity can be described as an outcome of the interactions between the urban form, the season climate, and the individual (Figure 1). The aim of this study is to test this proposition in the winter settlement of Luleå, Sweden, which experiences significant seasonal climate variation and is subject to rapid ongoing climate change [3,4]. This testing is problematized around the central research question; what attracts and hinders people from being outdoors in the built environment during the winter season and how can this knowledge underpin a model for soft-mobility?

For this paper, soft-mobility is defined as human-powered, non-motorized ways of getting around, such as walking, cycling, skating, or skiing, that have relatively little impact on the environment and require people to be physically active [5]. Promoting soft-mobility is considered to help deliver outcomes including social cohesion, resource efficiency, sustainability, and better land economy [6,7,8,9,10,11,12,13,14].

This examination is important because it looks to expand the discussion of the public realm to include the built environment, seasonal climate variation, and the individual. This allows for a more elaborate discussion about how urban design can better enable people to be active, outdoors, all year round, with the follow-on health benefits this can bring.

### 1.1. Policy Context

For urban design, today’s overarching policy context is the Sustainable Development Goals set out by the United Nations [15]. Because this study deals with soft-mobility in the built environments, the main body of the work falls under Goal 11, which is to “make cities and human settlements inclusive, safe, resilient and sustainable.” However, since the research looks at soft-mobility in relation to the human wellbeing benefits that can come from physical activity, it is also related to Goal 3 “to ensure healthy lives and promote wellbeing for all at all ages”. Additionally, as global warming is a focus of this study, it is linked to the need to take urgent action to combat climate change and its effects (Goal 13).

#### 1.1.1. Sustainable Development

The world commission on environment and development (WCED) report ‘Our Common Future’ [16] and the 1992 Rio Earth Summit introduced the concept of sustainable development, which UN members have adopted as an overarching policy. In urban design, the adoption of sustainable development as an overarching policy has seen a focus on urban forms that have a reduced impact on the environment. In many cases, these urban forms have focused on soft-mobility as a key design objective. Since the start of the 21st century, walkable communities—a prerequisite for soft-mobility —have been a mainstream part of urban planning. Importantly, urban forms that support mobility without reliance on motorized vehicles are seen as helping to reduce emissions and pollution [10], which is vital for efforts to slow down climate change.

#### 1.1.2. Climate Change

Global warming was first hypothesized (and disputed) in the 1930s, and is attributed to human activity by most researchers [17,18,19]. In brief, climate change is caused by the emission of pollutants known as greenhouse gases into the atmosphere; the magnitude of the change depends on the identity and quantity of the gases that are emitted. One factor determining emissions is land use and land cover [20]. Therefore, the planning, structure and form of our cities and their workings are determinants of gaseous emissions. The form of settlements and the mobility choices they facilitate influence transport emissions and therefore affect climate change.

This is important to this research because rising temperatures are likely to significantly influence people’s mobility choices. Awareness of the impact of climate change on soft-mobility is particularly important for winter communities like Luleå, Sweden. This is because over the last 50 years, Arctic temperatures have risen twice as quickly as the global average temperature [4,21]. Therefore, the effects of climate change are becoming apparent much more quickly in Arctic and sub-arctic regions.

#### 1.1.3. Human Health and Wellbeing

While urban design cannot be charged with delivering human health or wellbeing objectives, it can contribute by creating places that are attractive and safe for outdoor soft-mobility. Such places can enable people to be more physically active by using active modes of transport that increase their metabolic rates, enhance their physical capabilities, and confer health benefits. Seasonality in physical activity is a concern [22], wherefore opportunities for physical activity on a daily basis need to be created.

This is important, because recent decades have seen the emergence of links between non-communicable diseases, such as cancer, coronary heart disease, type 2 diabetes, and physical activity levels [23,24]. Consequently, there has been an increased emphasis on how the built environment can enable outdoor soft-mobility and activity. In particular, there is an interest in promoting built environments that enable active forms of mobility, such as walking or cycling [11,12]. 

## 2. Materials and Methods

This study builds upon wider research at Luleå University of Technology, Sweden into the design of attractive cold climate settlements. The data used in this study was originally collected to address a series of sub-research questions that underpin the main research question explored here: What attracts and hinders people from being outdoors in the built environment during the winter season and how can this knowledge underpin a new model for soft-mobility?

The sub-research questions and respective methods are outlined in Table 1, and full details of these separate studies can be found in articles by Chapman et al [5,25,26].

The research is grounded in a qualitative approach [27] in which empirical data was interpreted qualitatively. Because the research questions relate to how the interaction between the winter season and the urban environment creates connectivity for soft-mobility, the research is grounded in urban design. A case study location was selected strategically [28] based on the definition of a winter city as a city that experiences long snow-covered winters. The three methods (mental mapping, photo elicitation, and questionnaire) were all used to gather data of the same matter: how soft-mobility is perceived in the winter season. The results from the mental mapping focuses on the urban scale; how the residents perceive the urban structure, while the results from the questionnaire and the photo-elicitation explore how the public realm; streets and spaces of the neighborhood are altered in winter.

To address the aim of the main research question, results were integrated by triangulation [28]. Triangulation was used to crosscheck and validate the results obtained using the individual methods [29], and to characterize soft-mobility choices by different dimensions of the model. This yielded a more robust picture of connectivity for soft-mobility in the winter season. 

## 3. Results

### 3.1. Literature Studies

A systematic literature review of publications addressing the urban design of winter cities was undertaken. The review identified 35 documents covering various aspects of winter cities. The results were discussed in the context of rising outdoor temperatures due to climate change and their effect on winter weather conditions. The selected publications were also analyzed in relation to human wellbeing and how connectivity for soft-mobility can facilitate higher levels of physical activity and thus better health and wellbeing. The results were synthesized against key words related to the outdoor environment (distance, darkness, snow, rain, cold, wind, snow covered surfaces and ice-covered surfaces, and policy agendas (climate change, global warming, health and wellbeing, sustainability)

Several important conclusions were drawn from the literature review [25]. Publications established the importance of understanding seasonal climate variation as part of the urban design of winter settlements. The main factors considered in publications regarding the urban design of winter cities were solar access, wind defense, and snow management [30,31,32,33,34,35,36,37,38,39,40,41,42,43]. These urban design principles were mentioned in most of the reviewed publications, including those from Europe and North America (Figure 2). However, while publications were consistent in focusing on these three factors, little consideration was given to other conditions common to the winter season such as darkness, icy and ice covered surfaces and rain [25].

This study also showed that the number of publications dealing with urban design for winter cities peaked strongly between 1985 and 1989. This period broadly aligns with the founding and early years of the Winter Cities Association, Canada (WCA). This is important because while the WCA and Norman Pressman (a founding member) published their last book on winter cities in 2004, their publications are still some of the most widely available documents on the subject. 

The prominence of Pressman and the WCA in the discourse on winter cities is not a problem per se. However, there is a problem in that when their work ceased (in 2005 and 2004, respectively), research into winter cities slowed dramatically and may not have kept pace with problems that have since emerged. Notably, this review showed that most articles and books reviewed did not mention the term sustainability even though the concept was introduced by the report “Our common future” in 1987 [16]. All the literature was also reviewed against source information from the WHO on the health benefits of physical activity, an issue brought to prominence in 1996 by the U.S. Surgeon General’s report on physical activity and health [23]. In addition, the articles were reviewed against climate change scenarios, which became part of the international agenda in 1988 when the Intergovernmental Panel on Climate Change Secretariat (IPCC) was established. Most of the documents said nothing about the health benefits that the WHO relates to physical activity. Moreover, few addressed the changes in winter weather conditions that are resulting from climate change.

However, the documents did also highlight the importance of planning for the year-round outdoor environment of a winter city, rather than the brief summer period, and the importance of seeing the winter environment as a positive season and community asset. Here, texts highlighted the importance of capitalizing on the benefits of the winter season, such as opportunities for ice roads, winter markets, and lighting and the outdoor activities that winter affords, such as skiing and skating.

Whilst the literature studies were undertaken using appropriate keywords to find relevant articles, the searches only retrieved publications listed in established database, i.e., Scopus and Web of Science. Consequently, relevant works that were not in English or not listed in such databases, may have been overlooked.

### 3.2. Mental Mapping and Photo Elicitation

Empirical data was collected for this study using mental mapping exercises and photo elicitation. The results from both methods showed that seasonal climate variation could significantly alter the public realm and its usability.

In the built environment, the results from the mental mapping showed that the winter season could change a neighborhood’s local network of routes for soft-mobility, while the results from the photo elicitation exercise showed that in winter, build-ups of snow, ice, slush, and water reduced the area of the public realm usable for soft-mobility. In both cases, people highlighted that the winter season had a reductive effect on the street pattern and public realm. At the neighborhood scale, the mental mapping highlighted that the winter was seen to reduce the usable network of streets and spaces for soft-mobility. At the scale of the street, the photo elicitation exercise highlighted that the winter season was seen to reduce the space in the street available for soft-mobility (Figure 3 and Figure 4). Both were highlighted as major hindrances to soft-mobility in the winter season. The results from the photo elicitation exercise also established a ‘white-out’ effect of the winter season, which further hindered soft-mobility. Builds-up of snow and ice in winter made it difficult for people to identify the various elements of the streetscape (Figure 5), i.e., where the walkways and cycle lanes are, and where carriageways and parking for vehicles were. This ambiguity in the public realm was seen to hinder soft-mobility as vehicles were seen to dominate the space as it became undefined by winter.

Participants’ mental maps also differed noticeably with respect to the use of the sea. The summer maps of the water were free of interventions, but the winter maps showed that the frozen sea offered varied opportunities for soft-mobility. The mental mapping established that in winter, these areas offered new connections between places and varied opportunities for soft-mobility. Both formal, that is, maintained, and informal, i.e., non-maintained, pathways across the ice created seasonal connections between city districts and neighborhoods. These ephemeral pathways created new movement options around the city and supported a diverse range of soft-mobility choices, i.e., walking, cycling, skiing, skating, and kick-sled (Figure 5).

### 3.3. EAMQ-Climate Questionnaire

A questionnaire was developed to investigate people’s soft-mobility behaviors in relation to environmental condition in winter [5], by amending the Environmental Analysis of Mobility Questionnaire (EAMQ) [44]. The questionnaire was adapted to address the urban design of winter cities by focusing on the dimensions of distance, ambient conditions (dark, snow, rain), and terrain. It was expanded to address distance travelled by means of soft-mobility in both summer and winter, and a wider range of weather conditions experienced in winter. This made it possible to gather data on the effects of weather-related factors traditionally considered in winter city design (i.e., wind and snow) and the effects of other common weather conditions (i.e., ground surface properties of ice and snow). To differentiate the questionnaires, the suffix ‘Climate’ was added. 

The questionnaire consists of 18 items measured using a 5-step Likert scale, addressing distance and seven different ambient and terrain conditions; proposing either encountering or avoidance of the condition. The relative impact of each condition on soft-mobility in winter was analyzed, and considered as having a significant impact if exceeding the general elements effect on distance in winter. The results show that residents were slightly more reluctant to walk for more than a kilometer during winter (mean 2.5, SD 1.3) than during summer (mean 1.7, SD 1.0). 

Rainfall and icy surfaces were the top barriers [5]. Other important conditions were coldness, darkness, and wind. The conditions with the least impact on outdoor activity were snow and snow-covered surfaces (Table 2).

The responses to the questionnaire indicate that the traditional principles of winter city urban design only address some of today’s barriers to outdoor activity. While sun, wind, and snow management will clearly continue to be important, consideration needs also be given to other weather conditions such as ice, rain, and darkness, which also hinder soft-mobility.

Importantly, the questionnaire showed that icy surfaces and rain are now key considerations in winter. This is significant when reviewed in light of climate change at the case study location. In Norrbotten, the annual average temperatures have been rising since the early 1980s. Until then, the average temperature was around −2 °C. The average temperature now commonly fluctuates around 0 °C, and is predicted to rise over the coming years [3]. Here, the survey responses suggest today’s warmer winter season conditions of icy surfaces and rain, which are now more common because of climate change, are the conditions that are being experienced and influencing people soft-mobility choices in the community. 

## 4. Analysis and Discussion 

This research aimed to assess what attracts and hinders people from being outdoors in the built environment during the winter season, and how can this knowledge could underpin a new model for soft-mobility. An integrative model for soft-mobility was tested and nine key considerations were identified within the integrated dimensions of individual factors, urban form, and seasonal and climate change.

### 4.1. The Individual

While urban design is concerned with urban form, understanding how people move is a crucial aspect of public health science and physiotherapy. Enabling soft-mobility can increase levels of physical activity, which in turn increases people’s metabolic rates, physical capabilities, and general good health.

Central to this understanding is the concept of human motor control. Shumway-Cook and Woollacott [45] have created a theoretical framework for movement that links three interacting factors relevant to this research: the individual (and their capacities, perceptions, and experiences), the task (for example walking), and the environment. Central to Shumway-Cook and Woollacott’s theory is that these factors constrain human movement. They argue that sensory and perceptual aspects affect individuals, and that the environment is fundamental to understanding movement. 

Other central concepts are aiming to explain psychological processes involved in human behavior. One such theoretical framework is the theory of planned behavior by Ajzen [46], who argue that attitudes, subjective norms, and perceived behavioral control influence a person’s intention to perform a specific behavior (for example participate in leisure activities). Ajzen’s theory, explaining how the availability of opportunities and resources (such as, e.g., skills) and the perceived social pressure are significant for a person’s activity choice, is relevant to this research. 

This is important because urban design and health science theory concur that perceptions and understanding of the environment have important effects on human movement. However, while Shumway-Cook and Woollacott’s research majors on the individual and the task, their work is limited on the environmental dimension, which is the focus of this study.

For the individual, the results from this study show that:

#### 4.1.1. People’s Perceptions and Lived Experiences of the Outdoor Environment Are Changed by Seasonal Climate Conditions

This is a key consideration for the individual if we want to enable the potential health benefits of regular physical activity. Here, health guidance is given on a ‘regularity’ of activities, with levels typically reported on a daily or weekly basis [11,12]. Therefore, to enable these benefits settlement needs to be attractive for outdoor activity all year round and not just in the summer months [47,48,49].

#### 4.1.2. People’s Perceptions and Lived Experiences of Winter Barriers and Enablers to Soft-Mobility Are Evolving with Climate Change

The results from the EAMQ-Climate show that effects of climate change on outdoor soft mobility is being perceived and understood by the individual [5]. Importantly, it shows that climate change data highlighting global warming is now being experienced on the ground and this is altering people’s patterns of soft-mobility [48,50]. 

#### 4.1.3. New Climatic Conditions Challenges People’s Adaptive Strategies

Conditions of high impact (ice, rain) as perceived by the residents assessed by the EAMQ-Climate questionnaire and photo elicitation exercise [5] are noteworthy, as earlier research report air temperature, solar radiation, and wind speed as the main elements for people’s outdoor activity [51]. This highlights that the relative importance of climatic conditions, both real and perceived, are evolving, as people’s related behavior. Here, various factors, e.g., clothing and activity pattern, attitudes, experiences, and perceived control [46,51] influence place-related comfort and attendance.

### 4.2. Urban Form

For this study, the urban form is described as the built urban form of a place, and the public realm as the areas of the built environment that are there for everyone to see, use, and enjoy [52]. The results confirm that:

#### 4.2.1. The Winter Season Can Change a Place’s Public Realm but Excludes the Idea That It Can Alter the Urban Form of a Settlement Because the Built Form Stays the Same All Year Round

The results from the photo elicitation exercise highlight that the public realm in wintertime is in part an outcome of the interaction between the urban form and the winter conditions. This is important because it shows that the winter season can alter the form of the public realm and its usability, and thereby affect people’s perceptions and choices [26]. As the results for the mental mapping also show that winter season can change the spatial arrangement of the public realm and the relationship between buildings, the interaction between the built environment and winter season is an important consideration for urban design [32,41].

#### 4.2.2. The Interaction Between the Urban Form and Winter Season Alters a Settlement’s Physically Accessible Public Space

In summer, buildings and structures order the public space in the built environment [53]. However, the results from the mental mapping and photo elicitation exercise show that in winter, public space is physically changed by the covers of the winter season. Here, the public space is re-ordered by the urban form and the winter season together. As winter gathers in the urban form, this is likely to reduce the amount of physically accessible urban space, as the public realm becomes a type of container for the winter season (Figure 6). Again, this is an important consideration because the ‘built’ actual public space will not be the available space throughout the whole year [26].

#### 4.2.3. The Interactions Between the Urban Form and the Winter Season in the Built Environment Can Alter an Area’s Urban Grain, i.e., the Spatial Network That Defines an Area or Place.

This is important because the result from the mental mapping show that during the winter the networks of routes and pathways that make up an area’s spatial pattern [52] and arrangement can change (Figure 7). In the main, the winter season reduces the networks of routes and pathways. This implies that the interaction between the urban form and winter will produce a ‘coarser’ urban grain, reducing mobility options [25]. For planning, this is important because this interaction between the urban form and the winter season can alter area wider spatial arrangements, which is again an important consideration for urban design [54,55,56].

#### 4.2.4. The Winter Season Can Change the Visual Appearance of the Built Environment. in Particular, Build-Ups of Snow and Ice Can Alter Views and Vistas in a Neighborhood and the ‘White-Out’ Effect of the Winter Season Can Mask Elements of the Public Realm

The results from the photo elicitation exercise show that the interaction between the urban form and winter season can change the visual appearance and townscape of an area [25]. For the ‘townscape’ of an area, the winter season alters the way the components of a public realm are understood and how their combination is perceived, which is an important urban design consideration [7,57].

### 4.3. Climate and Climate Change

For seasonal climate variation and climate change, the results confirm that:

#### 4.3.1. While Many of the Processes That Shape the Urban Form Are the Outcomes of Human Activities in Winter Settlements, Such as Planning and Design, Seasonal Outcomes Are Partly Shaped by Natural Forces, in This Case, by the Seasonal Climate Variations

This result from the mental mapping, photo elicitation exercise and EAMQ-Climate is important because these non-human forces [58,59] still create identifiable patterns that are broadly repeatable over time. In this case, the interaction between the urban form and winter occur on a seasonal basis. As such, like man-made aspects of the built environment, these non-human interactions have a ‘rate of change’ [7]. However, this element has two rates of change. The first is part of the seasonal climate, which means interactions are annual, and the second is part of climate change, which with natural climate evolution is altering the winter season. As these two rates of change are connected, the patterns that are created by the interaction between the urban form and winter season are changing over time. Importantly, while these patterns are the result of natural forces, global warming can be linked to long-term human activity and in particular, it has been linked to town planning and land use [20].

#### 4.3.2. While the Documented Principles for Winter City Urban Design Major on Solar Access, Wind, and Snow Management, the Perceived Pallet of Metrological Conditions That Affect Soft-Mobility Is Much Wider

This is key finding from the literature studies and the EAMQ-Climate, as much of the existing guidance for the design of winter cities has been seen to be silent about climate change [25]. At the case study location, at least, the weather conditions identified as barriers to soft-mobility in winter can be linked to the warmer winter weather conditions attributed to climate change [5,25]. This is important as it shows that urban designers and planners cannot view the climate as static in design and need to see it as being in a constant state of evolution [26]. Methodologically, it means that survey instruments, such as EAMQ-Climate, may need to address a wider palette of conditions.

### 4.4. The Individual, Urban Form and Climate

As this analysis shows that the interaction between the urban form and the winter season can alter a settlement’s public realm, it follows that winter can change the public space. It has also shown that as winter gathers on the urban form, it is likely to reduce the amount of physically accessible urban space, both at the scale of the movement network and the street. As a result, it has been seen that this changes the connectivity and quality of the network of streets and pathways and alters how space is understood and used. These are all important influencing factor for people’s behavior [7,8,9] and link the three dimensions of the proposed soft-mobility model; the individual, urban form, and climate.

## 5. Conclusions

Taken together, this study has shown significant interactions between the individual (and their capacities, perceptions, and experiences); the urban form (the design of buildings, streets and spaces); and the seasonal climate (the winter season build-ups of snow, ice, rain, etc., and their management and maintenance) for soft-mobility.

The tested integrative model for soft-mobility provides a conceptual framework tool for the analysis and design of the urban realm, where nine key considerations were identified. These illustrate the importance of the urban form and the climate, and in places with significant seasonal climate variation, how seasonal interactions can reshape the public realm. In particular, it has shown that interactions can alter a place’s townscape, urban structure and urban grain, and streetscape, which are all important considerations for those involved in the production of the built environment.

It has also shown that people’s lived experiences of the outdoor environment are being altered by two rates of change: seasonal climate and on-going climate change. As such, new adaptive strategies will be required for safe and frequent soft-mobility choice in the public realm. 

For town planners and urban designers, this is important as it shows the need to identify and consider aspects within each one of the three-dimensions of soft-mobility; the individual, the urban form, and the seasonal climate/climate change. It is also important to acknowledge the significant interrelationships between the dimensions. Doing so will help develop schemes and proposals that are fit for the local context and community and support soft-mobility. In doing so, we will be better placed to help slow climate change by reducing energy consumption and emissions from motorized transport. It will also allow individuals to capitalize on the individual health opportunities soft-mobility can afford.

## Figures and Tables

**Figure 1 ijerph-16-03669-f001:**
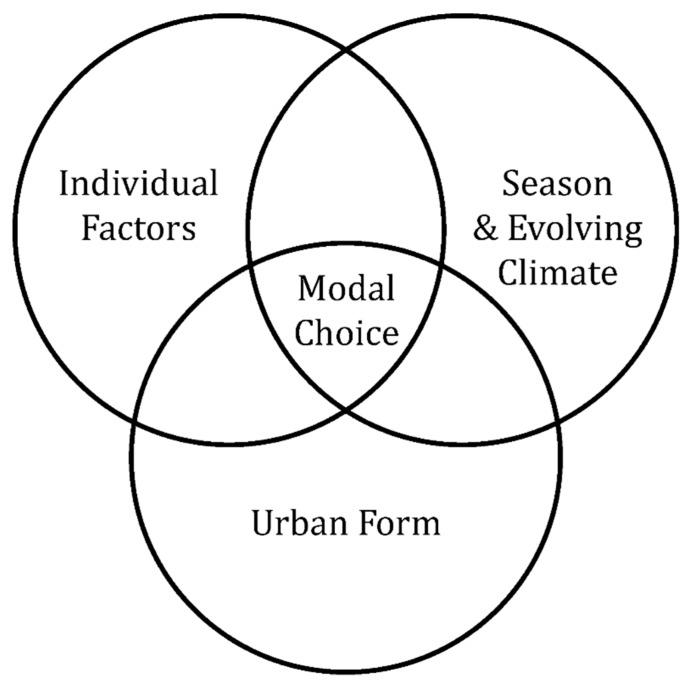
In the built environment, people’s mobility choices can be seen as an outcome of the interactions between the urban form, the seasonal conditions and climate change, and the individual.

**Figure 2 ijerph-16-03669-f002:**
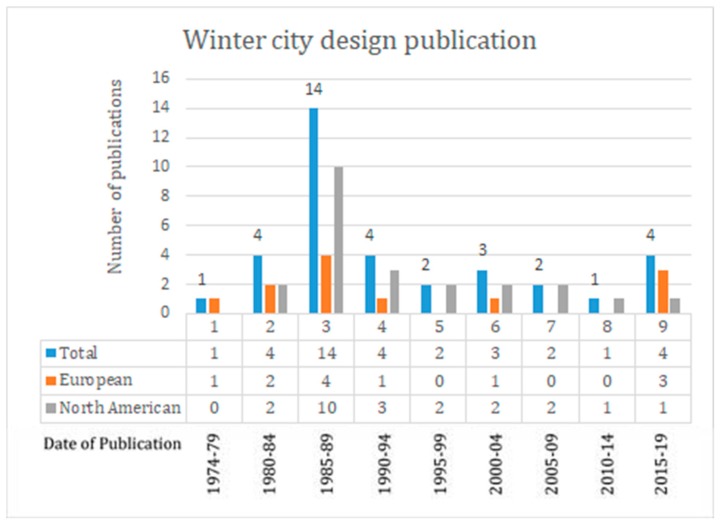
Chronology and geographical origin of the 35 publications included in the literature review.

**Figure 3 ijerph-16-03669-f003:**
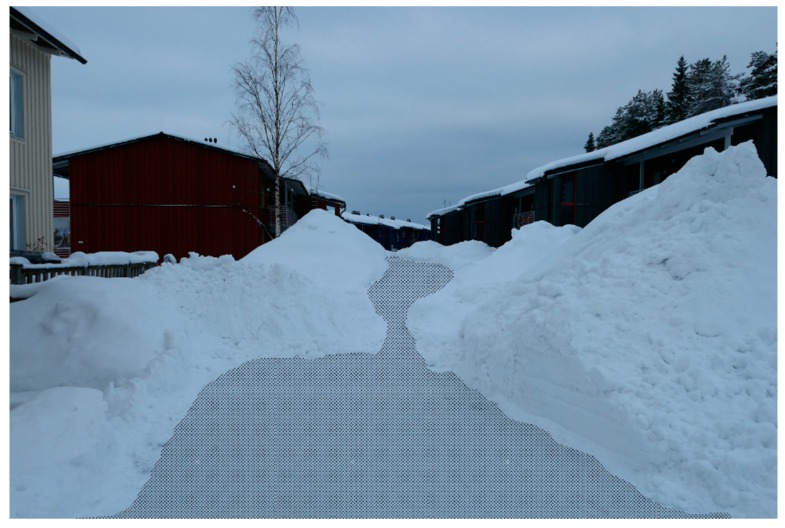
A residential street in Luleå in winter, with the usable area for soft-mobility indicated with the shading. These conditions can be in place for up to six months of the year.

**Figure 4 ijerph-16-03669-f004:**
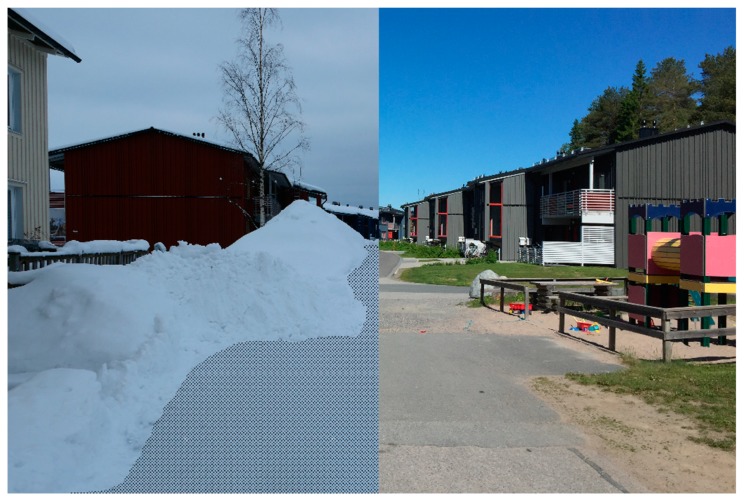
The same residential street as Figure 3, with the summer conditions shown on the right-hand side.

**Figure 5 ijerph-16-03669-f005:**
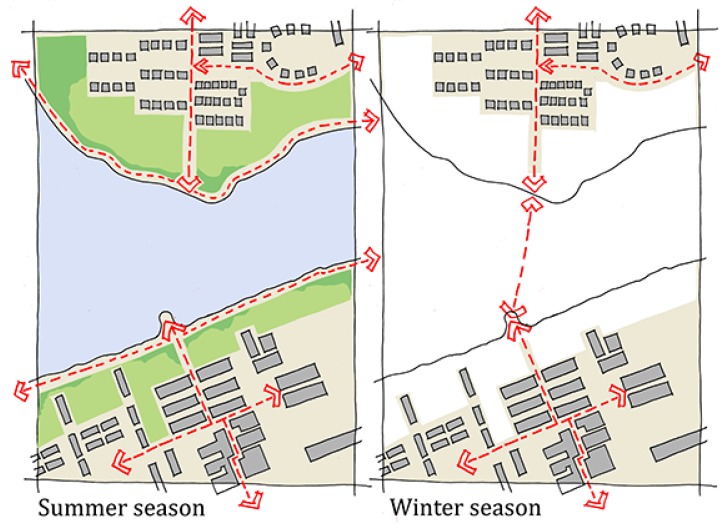
A sketch of how frozen water can enable new connections between neighborhoods.

**Figure 6 ijerph-16-03669-f006:**
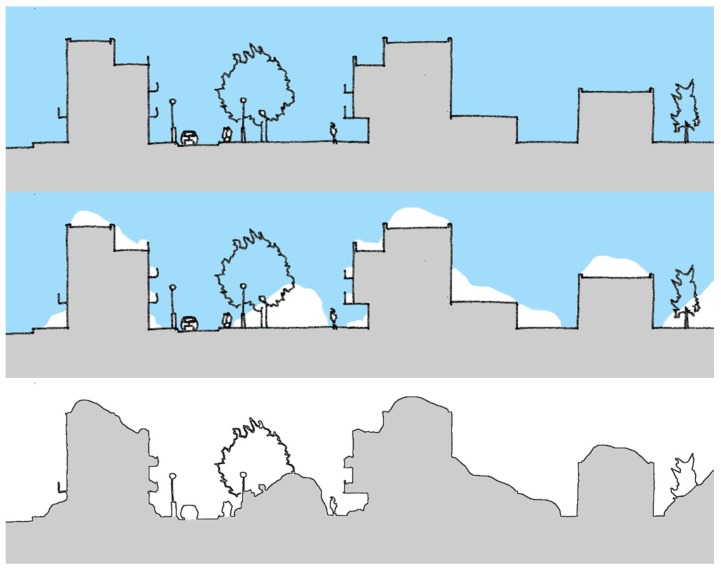
Illustration of how the winter season can re-order the space of the built environment. Top image: Section through the urban form of the street. Middle: Section of the street including winter season deposits. Bottom: The remaining space comprising the public realm in winter.

**Figure 7 ijerph-16-03669-f007:**
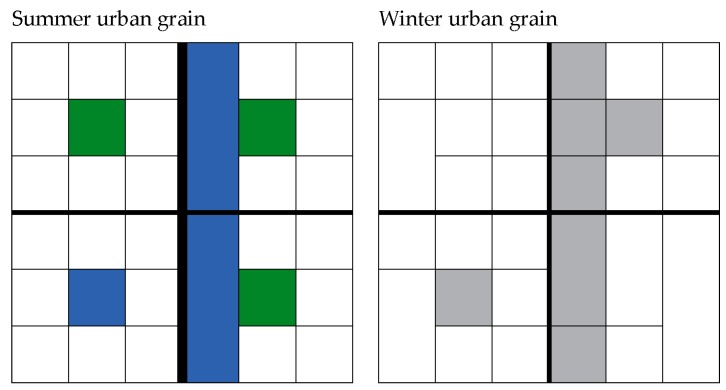
Diagrammatic illustration of how the interaction between the urban form and winter season were seen to be capable of altering an areas urban grain. Black lines indicate pathways, green areas are public spaces, and blue indicates water. The grey illustrates the white space created by winter.

**Table 1 ijerph-16-03669-t001:** Schedule of sub-research questions, methods, and materials.

Sub-Research Question	Type of Information Required	Methods	Participants/Materials
What is the current state of knowledge and practice relating to the urban design of winter cities? [25]	Academic books, peer-reviewed articles, design guides for winter settlements.	Literature Review: Documented principles of urban design for winter settlements.	35 (*n*) scholarly documents
What attractors, promoters, and hindrances to connectivity for soft mobility can be created in the public realm of winter cities? [5]	Information on how a neighbourhood’s perceived connectivity for soft-mobility can be altered by the winter season.	Mental mapping and photo elicitation: Mental maps of a case study neighbourhood in winter and summer.	15 (*n*) Residents
		Photographic images of barriers and enablers to connectivity for soft mobility in winter.	8 (*n*) Residents
What climate- and weather-based barriers and enablers to connectivity for soft mobility are created in the public realm of winter cities? [26]	Information on the perceived effects of the weather and winter season on soft-mobility.	EAMQ-Climate: Questionnaire based on which public realm and weather conditions inhibit or enable soft mobility.	212 (*n*) Residents

EAMQ = Environmental Analysis of Mobility Questionnaire. Amended version with suffix ‘Climate’.

**Table 2 ijerph-16-03669-t002:** Residents (*n* = 212) EAMQ-Climate ranking of the ambient and terrain barriers to soft-mobility in winter.

Ranking	Barrier Effect on Soft-Mobility	Mean (SD) Value ^1^	Main Winter Season Urban Design Consideration Highlight in Literature Study
1	Rain	3.0 (1.3)	No
2	Icy Surfaces	2.9 (1.4)	No
3	Coldness	2.5 (1.2)	No
4	Darkness	2.3 (1.3)	No
5	Wind	2.2 (1.2)	Yes
6	Snowing	2.2 (1.2)	Yes
7	Snow covered surfaces	2.0 (1.2)	Yes

^1^ Scale 1–5 (ranging from 1 = ‘never’ to 5 = ‘always’). Avoidance of walking more than 1 km in winter conditions (mean value 2.5) defined the cut of—point for significant conditions.

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
