# Peer review of "Toward an Integrated Model for Soft-Mobility"

_ijerph, 2019, doi:10.3390/ijerph16193669_

Round 1

Reviewer 1 Report

Thank you for the opportunity to review this research.  I found it to be an interesting topic and certainly in need of new research to inform design.  However, it needs a few major points to be addressed before it can be accepted for publication:

It needs an introduction to the concept of soft mobility.  Can you providea brief definition and key literature that is related to this study? Results for the literature review seem quite short.  If you reviewed 35 papers, are there any additional points or interesting results that could be highlighted?  Did you synthesize results through key words, themes, or location? Further details should be provided on the results of the mental mapping. How did this differ from the photo elicitation? The questionnaire needs further explanation and some statistical analysis to support the results.  For instance, what percentage of respondents ranked icy surfaces as the highest barrier? Are these results significant? Did it consist of only one question?   Analysis and discussion needs work on language for meaning and clarity.  Try to be very specific to relate the discussion points to specific and meaningful results from the research.  At the moment, it reads as a very brief set of points with only cursory links back to the research results.  Once the results are presented in greater detail, it should give further clarity to the discussion 

Other minor notes and suggestions are in the attached file.  

Author Response

1.  It needs an introduction to the concept of soft mobility.  Can you provide a brief definition and key literature that is related to this study?

Responses.  A brief definition of soft-mobility has been added and key literature related to soft-mobility has been included both urban design and health.

2.  Results for the literature review seem quite short.  If you reviewed 35 papers, are there any additional points or interesting results that could be highlighted?  Did you synthesize results through key words, themes, or location?

Responses.  A more detailed explanation of the method used and the synthesize of the results has been added. Some wider additional results are now included.

3.  Further details should be provided on the results of the mental mapping. How did this differ from the photo elicitation?

Responses.  Further details of the mental mapping have been added. Results from the mental mapping and photo elicitation have been differentiated.

4.  The questionnaire needs further explanation and some statistical analysis to support the results.  For instance, what percentage of respondents ranked icy surfaces as the highest barrier?

Responses.  Are these results significant? Did it consist of only one question? More information about the questionnaire and analysis of the results have been added Also mean values and cut off points for significance has been added in table 2. 

5.  Analysis and discussion needs work on language for meaning and clarity.  Try to be very specific to relate the discussion points to specific and meaningful results from the research.  At the moment, it reads as a very brief set of points with only cursory links back to the research results.

Responses.  The discussion points have now been linked directly to the methods and specific results. Additional text and illustrations have also been added to clarify the analysis.

6.  Once the results are presented in greater detail, it should give further clarity to the discussion

Responses.  Clearer links between results and discussion have been added.

7. Other minor notes and suggestions are in the attached file.

Responses.  All minor comments/ suggestions highlighted in the reviewers PDF document have been addressed.

Reviewer 2 Report

In line 29, the following reference is recommended to be added.

Bin Yang, Thomas Olofsson, Gireesh Nair, Alan Kabanshi. 2017. Outdoor thermal comfort and human behavior pattern under subarctic climate of north Sweden - a pilot study in Umeå. Sustainable Cities and Society, 28, 387-397.

In line 116, any publications outside Europe and North America? In line 156, why rain not snow, perhaps better call sleet? In line 168, change O to 0.

Author Response

In line 29, the following reference is recommended to be added. Bin Yang, Thomas Olofsson, Gireesh Nair, Alan Kabanshi. 2017. Outdoor thermal comfort and human behavior pattern under subarctic climate of north Sweden - a pilot study in Umeå. Sustainable Cities and Society, 28, 387-397.

Reference added.

In line 116, any publications outside Europe and North America?

A method state has been added highlighting that only document written in English were reviewed and this limited the geographical spread of documents included.

In line 156, why rain not snow, perhaps better call sleet?

Rain was highlighted by participants as a top barrier to soft-mobility. Snow was not. The survey did not include sleet, but this could be included in future surveys.

In line 168, change O to 0.

Corrected